# Community Structures and Antifungal Activity of Root-Associated Endophytic Actinobacteria in Healthy and Diseased Cucumber Plants and *Streptomyces* sp. HAAG3-15 as a Promising Biocontrol Agent

**DOI:** 10.3390/microorganisms8020236

**Published:** 2020-02-10

**Authors:** Peng Cao, Chenxu Li, Han Wang, Zhiyin Yu, Xi Xu, Xiangjing Wang, Junwei Zhao, Wensheng Xiang

**Affiliations:** 1Key Laboratory of Agricultural Microbiology of Heilongjiang Province, Northeast Agricultural University, No. 600 Changjiang Road, Xiangfang District, Harbin 150030, China; cp511@126.com (P.C.); licx95@126.com (C.L.); wanghan507555536@gmail.com (H.W.); yuzhiyin123@foxmail.com (Z.Y.); xuxi1758899581@163.com (X.X.); wangneau2013@163.com (X.W.); 2State Key Laboratory for Biology of Plant Diseases and Insect Pests, Institute of Plant Protection, Chinese Academy of Agricultural Sciences, Beijing 100193, China

**Keywords:** *Fusarium oxysporum* f. sp. *cucumerinum*, endophytic actinomycetes, HAAG3-15, biocontrol agent

## Abstract

Microorganisms related to plant roots are vital for plant growth and health and considered to be the second genome of the plant. When the plant is attacked by plant pathogens, the diversity and community structure of plant-associated microbes might be changed. The goal of this study is to characterize differences in root-associated endophytic actinobacterial community composition and antifungal activity between *Fusarium* wilt diseased and healthy cucumber and screen actinobacteria for potential biological control of *Fusarium* wilt of cucumber. In the present research, three healthy plants (also termed “islands”) and three obviously diseased plants (naturally infected by *F. oxysporum* f. sp. *cucumerinum*) nearby the islands collected from the cucumber continuous cropping greenhouse were chosen as samples. Results of culture-independent and culture-dependent analysis demonstrated that actinomycetes in the healthy roots were significantly more abundant than those of diseased roots. Moreover, there were seven strains with antifungal activity against *F. oxysporum* f. sp. *cucumerinum* in healthy cucumber roots, but only one strain in diseased cucumber roots. Out of these eight strains, the isolate HAAG3-15 was found to be best as it had the strongest antifungal activity against *F. oxysporum* f. sp. *cucumerinum*, and also exhibited broad-spectrum antifungal activity. Thus, strain HAAG3-15 was selected for studying its biocontrol efficacy under greenhouse conditions. The results suggested that the disease incidence and disease severity indices of cucumber *Fusarium* wilt greatly decreased (*p* < 0.05) while the height and shoot fresh weight of cucumber significantly increased (*p* < 0.05) after inoculating strain HAAG3-15. On the basis of morphological characteristics, physiological and biochemical properties and 100% 16S ribosomal RNA (rRNA) gene sequence similarity with *Streptomyces sporoclivatus* NBRC 100767^T^, the isolate was assigned to the genus *Streptomyces*. Moreover, azalomycin B was isolated and identified as the bioactive compound of strain HAAG3-15 based on analysis of spectra using a bioactivity-guided method. The stronger antifungal activity against *F. oxysporum* f. sp. *cucumerinum*, the obvious effect on disease prevention and growth promotion on cucumber seedlings in the greenhouse assay, and the excellent broad-spectrum antifungal activities suggest that strain HAAG3-15 could be developed as a potential biocontrol agent against *F. oxysporum* f. sp. *cucumerinum* used in organic agriculture. These results suggested that the healthy root nearby the infected plant is a good source for isolating biocontrol and plant growth-promoting endophytes.

## 1. Introduction

Cucumber (*Cucumis sativus* L.), belonging to the family Cucurbitaceae [1], is a very important vegetable which possesses remarkable economic and dietary value. It has been around for over three thousand years as a monoecious annual cultivable plant [2,3]. In addition, cucumber is well known for its softness and succulence and contains a variety of nutrients, such as potassium, copper, manganese, phosphorus, pantothenic acid, dietary fibers, and vitamins (A, C, K, and B6) [4].

However, cucumber is susceptible to many pathogens and pests [5]. Cucumber *Fusarium* wilt, induced by the pathogen *Fusarium oxysporum* f. sp. *cucumerinum*, is a typical soil-borne fungal disease and also one of the most important cucumber diseases in worldwide [6,7]. The disease could reduce ~10% to 30% of cucumber production and cause quality degradation, which results in serious economic losses [8,9]. *Fusarium* wilt of cucumber disease may appear throughout the whole growth period of cucumber plant, and the disease incidence at early stages is more serious [10,11]. The symptoms of the disease are vascular and root wilt which eventually cause plant death [11,12]. Chemical control agents are implicated in ecological, environmental, and human health problems, and pathogens can develop resistance to them [13]. Traditional ways of crop rotation and seeding grafting could be applicable for controlling *Fusarium* wilt; however, these methods are high-cost and laborious [14]. Moreover, soil fumigation is also an efficient approach to control the extension of soil-borne disease, but this strategy is labor-consuming and inconvenient, which limits its application [9].

Up to now, agricultural scientists paid much attention to an efficient, environmentally friendly, and sustainable method, biocontrol, which is used for protecting plants against soil-borne diseases [15]. A number of antagonistic microbes were investigated and studied to control various plant pathogens, such as *Pseudomonas* spp., *Bacillus* spp., *Streptomyces* spp., *Trichoderma* spp., and *Paenibacillus* spp [13,16,17,18,19,20,21,22]. Kareem et al. reported that *Trichoderma longibrachiatum* NGJ167 could be used to control *Fusarium* wilt of cucumber [13], *Bacillus subtilis* 9407 was recorded as a biocontrol agent against bacterial fruit blotch of melon [20], and *Pseudomonas aeruginosa* BRp3 could reportedly be applied in controlling bacterial leaf blight of rice [22]. *Streptomyces* genus is one of the most efficient groups, with the capacity of preventing plant fungal diseases [17,19,21,23,24]. *S. exfoliates* FT05W and *S. cyaneus* ZEA17 were documented as biocontrol agents against lettuce drop caused by *Sclerotinia sclerotiorum* [19], *S. griseochromogenes* and *S. lydicus* WYEC108 were reported as biocontrol agents against the rice blast fungus, *Magnaporthe oryzae* (*Pyricularia oryzae*) [21], and *S. albospinu*s CT205 was found to have biocontrol potential against cucumber *Fusarium* wilt [24]. *Streptomyces* with efficient rhizosphere and/or the inner regions of plant tissue colonization could prevent fungal pathogens and promote plant growth by inoculating spore suspensions on seeds or seedlings [19,25,26,27]. The genus *Streptomyes* is known for its capacity of producing abundant secondary metabolites with bioactivities against plant pathogens [28]. Thus, the isolation of *Streptomyces* with antibacterial activity is recognized as a crucial strategy in the prevention and control of plant diseases and development of agriculture, as well as ecosystem safety.

Endophytes which can reside within the plants through parasitic, symbiotic, or mutualistic modes without inducing apparent infections or symptoms of disease for the whole or part of their life history [29,30] represent a portion of the microbes associated with plant. The phylum *Actinobacteria* was reported as the major portion of endophytic microbes, while *Streptomyces* was reported as the main content of endophytic actinomycetes in most plants [31,32,33,34,35,36,37,38,39,40,41,42]. It was also reported that many endophytic actinobacteria could control plant pathogens, improve plant stress resistance, and promote plant growth [43,44,45,46]. Therefore, the application of endophytic biotcontrol agents is the current research hotspot [47] and could provide biocontrol strategies in future.

Microorganisms related to plant roots are vital for plant growth and health and considered to be the second genome of the plant. When the plant is attacked by plant-pathogenic microorganisms, the diversity and community structure of plant-associated microbes may be changed [48,49]. The goal of this study is to characterize differences in root-associated endophytic actinobacterial community composition and antifungal activity between *Fusarium* wilt diseased and healthy cucumber, as well as to screen actinobacteria for potential biological control of *Fusarium* wilt of cucumber. In our present research, three healthy plants (also termed “islands”) [50] and three obviously diseased plants nearby the islands collected from a cucumber continuous cropping greenhouse were chosen as samples for culture-independent and culture-dependent analysis. A preliminary study of antifungal activities of these strains against *F. oxysporum* f. sp. *cucumerinum* was performed and compared. Strain HAAG3-15 with the strongest antifungal activity was selected for investigating its biocontrol effect on potted plants. Furthermore, the bioactive constituent with antifungal activity of strain HAAG3-15 was isolated and the chemical identity was determined. This would be of high importance for the source of antagonistic strains and biocontrol of cucumber *Fusarium* wilt, as well as other plant fungal diseases.

## 2. Materials and Methods

### 2.1. Sampling of Healthy and Diseased Plants

In June 2017, three healthy cucumber plants (also termed “islands”) and three obviously diseased cucumber plants (infected by *F. oxysporum* f. sp. *cucumerinum*) nearby the islands collected from the cucumber continuous cropping greenhouse (320 m^2^, plastic film) of Northeast Agricultural University, Heilongjiang province, northeast China (45°41′ north (N), 126°37′ east (E)) were chosen as samples. Each group included one healthy plant and one obviously diseased plant nearby the island. These three groups were named H1-D1, H2-D2, and H3-D3. All cucumber plants surveyed in the current study were in the initial flowering stage of cucumber.

### 2.2. DNA Extraction, Sequencing, and Data Analysis

The loose soil attached to the cucumber roots was firstly removed by gentle shaking. Then, the roots were washed in water with an ultrasonic step (160 W, 15 min) to thoroughly clean off surface soils and adherent epiphytes. Then, the total DNA was isolated from the roots using the Cetyltrimethylammonium Ammonium Bromide (CTAB) method. The purity of DNA was checked using 1% agarose gels, and DNA concentration was determined with a NanoPhotometer spectrophotometer (Implen, München, Germany). The V3–V4 regions of 16S ribosomal RNA (rRNA) genes of bacterial DNA were amplified using the primer pair 341F (forward; 5′–CCTAYGGGRBGCASCAG–3′) and 806R (reverse; 5′–GGACTACNNGGGTATCTAAT–3′) with the barcode and sequenced on an Ion S5^TM^ XL platform at Beijing Novogene Technology Co. Ltd. (Beijing, China), generating 600 bp single-end reads. The raw data were filtered (removing low-quality reads less than 17) according to the Cutadapt (V1.9.1, http://cutadapt.readthedocs.io/en/stable/) [51] quality control process. The reads were compared with Silva database (https://www.arb-silva.de/) [52] using the UCHIME algorithm (http://www.drive5.com/usearch/manual/uchime_algo.html) [53], and then the chimera sequences [54] were removed to obtain clean data. The Uparse software (v7.0.1001, http://drive5.com/uparse/) [55] was used for sequences analysis. All sequences were clustered into operational taxonomic units (OTUs) on the basis of a sequence similarity of ≥97%, and the representative sequence was annotated with taxonomic information based on the Silva Database (https://www.arb-silva.de/) [52] using the Mothur algorithm. Determination of the difference in dominant species among different samples (groups) and multiple sequence alignment were carried out using the MUSCLE software (Version 3.8.31, http://www.drive5.com/muscle/) [56]. Observed species and Chao 1 were calculated with QIIME (Version1.9.1, http://qiime.org), and principal component analysis (PCA) was used for analyzing the ordinations of community patterns.

### 2.3. Isolation and Maintenance of Endophytic Actinomycetes

The cucumber root samples were air dried for 24 h at room temperature and weighed. The roots were cut into pieces of 5–10 mm in length and then subjected to a seven-step surface sterilization procedure [57]. The samples were then ground with a sterile mortar and pestle, employing 1 mL of 0.5 M phosphate buffer saline (pH 7.0) per 100 mg tissue. Tissue particles were allowed to settle down at 4 °C for 20–30 min. The suspensions of each sample were all spread on plates of humic acid–vitamin agar [58], Gause’s synthetic agar No. 1 [59], dulcitol–proline agar [57], cellulose–proline agar [57], and arginine–alanine–granulose agar [57], supplemented with cycloheximide (50 mg∙L^−1^) and nalidixic acid (20 mg∙L^−1^). Endophytic strains were incubated at 28 °C until single colonies were observed. Single actinomycete colonies growing on the plates were isolated and purified on oatmeal agar (International *Streptomyces* Project medium 3, ISP 3) [60]. The isolates were prepared on ISP 3 medium and kept at −80 °C (under 30% glycerol) for long-term storage and at 4 °C as source cultures.

### 2.4. Screening the Isolates with Antifungal Activity

To screen antagonistic actinomycetes, the antifungal activity of these isolates was determined against the pathogen *F. oxysporum* f. sp. *cucumerinum*. These isolates were streaked on ISP 3 medium and cultivated for seven days (five repetitions). The pathogenic fungus (*F. oxysporum* f. sp. *cucumerinum*) was cultured on potato dextrose agar (PDA) for one week [61]. Mycelia discs (6 mm diameter) of each pathogen were picked up and put in the center of different plates which contained freshly prepared PDA medium, and the strains were point-inoculated at the margin areas which were 3 cm away from the central pathogen colony using an inoculating needle, and then cultured in an incubator at 28 °C for seven days. All the experiments above were repeated three times. The inhibition of fungal growth on each plate was calculated as described below [62]. The antifungal activity of antagonistic strains against other nine pathogenic fungi (*Corynespora cassiicola*, *Setosphaeriaturcica turcicaf*, *Colletotrichum orbiculare*, *Alternaria solani*, *Helminthosporium maydis*, *Sphacelotheca reiliana*, *Sclerotinia sclerotiorum*, *Phytophthora sojae*, *Rhizoctonia solani*) was also determined as described in Equation (1).
(1)$$\mathrm{Inhibition}\;\mathrm{of}\;\mathrm{fungal}\;\mathrm{mycelial}\;\mathrm{growth}\;(\%)\;=\;\frac{A-B}{A}\;\times \;100,$$
where A is the mycelial growth of fungal pathogen in the absence of antagonists, and B is the mycelial growth of fungal pathogen in the presence of antagonists.

### 2.5. Morphological and Physiological Characterization

Gram staining was performed using a standard method. Morphological characteristics of strain HAAG3-15 were observed by light microscopy (Nikon ECLIPSE E200, Nikon Corporation, Tokyo, Japan) and scanning electron microscopy (Hitachi SU8010, Hitachi Co., Tokyo, Japan) using cultures grown on ISP 3 medium at 28 °C for three weeks; samples for scanning electron microscopy were prepared as described by Jin et al. [63].

### 2.6. Genomic and Phylogenetic Analysis

For DNA extraction, strain HAAG3-15 was cultured in ISP 2 medium at 28 °C for four days, and then the cultures were centrifuged to harvest the cells. Genomic DNA extraction was carried out using a TIANamp Bacteria DNA Kit (TIANGEN Biotech, Co., Ltd., Beijing, China). PCR amplification of the 16S rRNA gene was performed using a standard procedure [64]. PCR products were purified and cloned into the vector pMD19-T (Takara Bio Inc., Dalian, China) and sequenced using an Applied Biosystems DNA sequencer (model 3730XL, Applied Biosystems Inc., Foster City, CA, USA). The almost full-length 16S rRNA gene sequence of strain HAAG3-15 (1519 bp) was obtained and submitted to the EzBioCloud server (https://www.ezbiocloud.net/) for comparison with type strains [65], retrieved using NCBI BLAST (https://blast.ncbi.nlm.nih.gov/Blast.cgi;), and then submitted to the GenBank database. The phylogenetic tree was built based on the 16S rRNA gene sequences of strain HAAG3-15 and related reference species. Sequences were multiply aligned in Molecular Evolutionary Genetics Analysis (MEGA) software version MEGA7.0 using Clustal W algorithm and manually modified if necessary. The phylogenetic tree was constructed using the neighbor-joining [66] algorithm with MEGA [67]. A bootstrap method with 1000 replicates was used to evaluate the stability of the topology of the phylogenetic tree [68]. Kimura’s two-parameter model was used for generating a distance matrix [69]. All positions in the dataset containing gaps and missing data were deleted (complete deletion option).

### 2.7. Greenhouse Biocontrol Assay Using HAAG3-15

The capacity of strain HAAG3-15 to control cucumber *Fusarium* wilt and promote the growth of cucumbers (Jinyan four varieties) was evaluated using a pot experiment with four treatments (F, F + S, N, and S) under greenhouse conditions. The soil used in the present study was steam-sterilized three times (121 °C, 30 min). In the treatment F, the spore suspension of *F. oxysporum* f. sp. *cucumerinum* (2 mL, 4–5 × 10^4^ colony-forming units (CFU)∙mL^−1^) was irrigated in the soil while cucumber was transplanted. In the treatment F + S, the spore suspension of isolate HAAG3-15 (2 mL, 4–5 × 10^6^ CFU∙mL^−1^), together with that of *F. oxysporum* (2 mL, 4–5 × 10^4^ CFU∙mL^−1^), was irrigated in the soil while the cucumber was transplanted. In the treatment N, no microbial suspension was added to the soil. For the treatment S, the spore suspension of isolate HAAG 3-15 (2 mL, 4–5 × 10^6^ CFU∙mL^−1^) was irrigated in the soil when the cucumber was transplanted. For each treatment, 30 two-week old cucumber seedlings were used and cultivated in plastic pots (15 cm diameter, one cucumber seedling per pot). This study was performed under greenhouse conditions with average temperature of 25 °C, relative humidity of about 60%, and 12 h of illumination (11.8 W/m^2^) per day. The cucumber seedlings were watered every two days and no fertilizers were used. Fifteen cucumber seedlings randomly harvested from the pots of each treatment were used to measure their shoot fresh weights and heights after cultivating after four weeks. The disease symptoms of all cucumber seedlings per treatment were investigated in this study. Severity of disease symptoms was recorded using an index ranging from 0 (healthy plant) to 4 (dead plant). The plant disease index (DI) was calculated according to the following formula: DI = [∑ (Ni × i)/ (N × 4)] × 100, where i means a 0–4 disease level, and Ni means the plant number of reaction i [70].

### 2.8. Isolation and Characterization of the Antifungal Compound

The antifungal compound was separated based on the antifungal (*F. oxysporum* f. sp. *cucumerinum*) activity-guided method from the extraction. Strain HAAG3-15 was inoculated into 250-mL Erlenmeyer flasks filled with 50 mL of sterile tryptic soy broth (seed medium, TSB, Beijing AOBOXING BIO-TECH CO., LTD., Beijing, China) and cultured for two days at 28 °C. Then, the seed culture (12.5 mL) was transferred into 1000-mL Erlenmeyer flasks containing 250 mL of production medium (soybean flour 20 g, peptone 2 g, glucose 20 g, soluble starch 5 g, yeast extract 2 g, NaCl 4 g, K_2_HPO_4_ 3H_2_O 0.5 g, MgSO_4_ 7H_2_O 0.5 g, CaCO_3_ 2 g, and distilled water 1 L; pH 7.2–7.4) and incubated at 28 °C for seven days on a rotary shaker (250 rpm). Next, 50-L cultures were obtained and filtered, and 30 L of mycelial cake was harvested. Then, the mycelial cake was washed with 3 L of distilled water and extracted with 3 L of methyl alcohol. The supernatant and wash water were subjected to a Diaion HP-20 resin column (500 mm × 100 mm inner diameter (i.d.)) eluting with 95% EtOH (5 L). The MeOH extract and the EtOH eluents were evaporated under reduced pressure to 1 L at 50 °C, and the resulting concentrate was extracted three times using EtOAc (5 L) and then concentrated to yield a residue (22 g) in the same conditions. The crude extract was resolved by a silica gel (100–200 mesh) column eluted with a stepwise gradient of CHCl_3_/MeOH mixtures with growing polarity (100:0–50:50, *v*/*v*) to obtain three fractions (Fr. 1–3) based on the Thin-Layer Chromatography (TLC) profiles, performed with a solvent system of CHCl_3_/MeOH (9:1). Fr. 2 showed antifungal activity and was further purified by a Sephadex LH-20 gel column eluted with CHCl_3_/MeOH (1:1, *v*/*v*), giving two fractions Fr. 2-1 and Fr. 2-2, referring to the TLC profiles. Fr. 2-1 showed antifungal activity and was further separated by semi-preparative HPLC eluting with a CH_3_CN–H_2_O mixture (48:52, *v*/*v*) using a reversed-phase column (Zorbax SB-C18, 5 mm, 250 × 9.4 mm inner diameter) to obtain compound 1 (*t*_R_ 27.0 min, 12 mg). The eluates were detected by a photodiode array detector at 254 nm with a flow rate of 1.5 mL/min at 25 °C. 

Spectroscopic analysis was used to determine the structure of the antifungal compound. ^1^H and ^13^C NMR spectra were measured with a Bruker DRX-600 (600 MHz for ^1^H and 150 MHz for ^13^C) spectrometer (Bruker, Rheinstetten, Germany). Electrospray ionization (ESI) MS data were obtained using an Agilent G6230 Q-TOF mass instrument (Agilent Corp., Santa Clara, CA, USA).

### 2.9. Statistical Analysis

The data were analyzed using analysis of variance (ANOVA) followed by Duncan’s multiple-range test (*p* ≤ 0.05) using statistical software SPSS version 17.0 (SPSS Inc., Chicago, IL, USA). The results were expressed as means ± SD.

## 3. Results

### 3.1. Culture-Independent Communities

In the current study, a total of 300,923 high-quality reads classified as 8708 OTUs from the microbiome in cucumber root were determined. The raw sequencing reads were deposited to NCBI SRA (National Center for Biotechnology Information Short Read Archive) for this project under accession numbers SRR10589211–SRR10589216. The predominant bacterial phyla of healthy and diseased cucumber roots were all Proteobacteria, Actinobacteria, and Bacteroidetes, but the relative abundance of phylum Actinobacteria in the healthy samples was more significant (*p* < 0.05) than in diseased samples (Figure 1). Including *Streptomyces*, the relative abundance of genera in Actinobacteria (top 30) in the healthy samples was greater than in diseased samples (Appendix A), except for *Sporichthya* (Appendix A). The bacteria α-diversity Chao 1 index of the healthy root was significantly higher than that of the diseased root (Figure 2). Principal component analysis (PCA) suggested a clear difference between the bacterial community (Analysis of similarities (ANOSIM) for bacteria, *p* = 0.1) of healthy and diseased cucumber roots (Figure 3). The principal component analysis (PCA) explained 34% and 18% of the variation in the bacterial communities.

### 3.2. Isolation of Endophytic Actinomycetes

A total of 263 endophytic actinomycetes colonies were isolated from healthy and diseased cucumber roots. Out of this number, 50 (58.1%), 66 (75.8%), and 57 (63.3%) isolates originated from the roots of healthy cucumber in Group 1 (H1-D1), Group 2 (H2-D2), and Group 3 (H3-D3), respectively, whereas 36 (41.9%), 21 (24.2%), and 33 (36.7%) isolates originated from the roots of diseased cucumber in all three groups. The colony-forming units (CFU) per gram of root varied widely among healthy and diseased plants, while also indicating that endophytic actinomycetes in healthy cucumber roots were more significantly (*p* < 0.05) abundant than in diseased cucumber roots (Figure 4).

### 3.3. In Vitro Antagonistic Activity Assays

There were eight strains showing antagonism to *F. oxysporum* f. sp*. cucumerinum*. Among these eight antagonistic strains, seven strains (HGS1-1, HGS2-18, HGS3-17, HAAG3-4, HAAG3-8, HCPA2-26, and HAAG3-15) were isolated from healthy cucumber roots, while only one strain (DCPA1-15) was isolated from diseased cucumber roots. Strain HAAG3-15 isolated from the root of healthy cucumber showed 71% inhibition of mycelial growth (Appendix A), whereas the other seven strains only exhibited 17% to 45% inhibition. In addition, strain HAAG3-15 also exhibited stronger antifungal activities against other nine pathogenic fungi (*Corynespora cassiicola*, *Setosphaeriaturcica turcicaf*, *Colletotrichum orbiculare*, *Alternaria solani*, *Helminthosporium maydis*, *Sphacelotheca reiliana*, *Sclerotinia sclerotiorum*, *Phytophthora sojae*, and *Rhizoctonia solani*) than other antagonistic strains (Figure 5).

### 3.4. Characterization and Identification of the Isolate HAAG3-15

The morphological characteristics of strain HAAG3-15 showed that it belonged to the genus *Streptomyces* [71]. The strain formed well-developed, branched substrate hyphae and aerial mycelium that differentiated into straight or flexuous spore chains consisting of cylindrical spores (0.55–0.81 μm × 0.75–1.22 μm), and the spore surface was rough (Figure 6) after cultivation for three weeks. The strain exhibited good growth on all tested media. Diffusible pigments were not observed on any of the media used in this study for strain HAAG3-15. The isolate was observed to grow in a pH range of 6.0–8.0 (optimum pH 7.0) and 10–40 °C (optimum 28 °C), as well as in the presence of 0%–2% NaCl (*w*/*v*, optimally 0%).

The almost-full length 16S rRNA gene sequence (1519 bp) of strain HAAG3-15 was deposited as MN726931 in the GenBank/EMBL/DDBJ (European Molecular Biology Laboratory/DNA Data Bank of Japan) databases. Based on EzBioCloud analysis, strain HAAG3-15 belongs to the genus *Streptomyces* and is most closely related to *Streptomyces sporoclivatus* NBRC 100767^T^ (100% identity) and *S. antimycoticus* NBRC 12839^T^ (100%). Phylogenetic analysis based on 16S rRNA gene sequences with the neighbor-joining tree suggested that the strain clustered within the genus *Streptomyces* and formed a stable subclade with *S. sporoclivatus* NBRC 100767^T^ and *S. antimycoticus* NBRC 12839^T^ (Figure 7).

### 3.5. Greenhouse Biocontrol Assay of Strain HAAG3-15

In the experiment under greenhouse conditions, the disease index and incidence, shoot fresh weight, and height of cucumber seedlings were measured upon transplanting after four weeks; the average values were calculated, and the results are presenting in Table 1. After two weeks of the inoculation of *F. oxysporum* f. sp*. cucumerinum*, some visual external wilt symptoms (yellowing of leaves and stems) of cucumber seedlings were exhibited in both F (inoculated only *F. oxysporum* f. sp*. cucumerinum*) and F + S (inoculated both HAAG3-15 and *F. oxysporum* f. sp*. cucumerinum*), and the typical symptom of chlorosis emerged and spread from older leaves to younger leaves. With the application of strain HAAG3-15, the treatment F + S could significantly reduce the disease severity of *F. oxysporum* f. sp*. cucumerinum* on cucumber seedlings after transplanting at four weeks (Figure 8). For F + S treatment, only 10 (30%) infected plantlets showed typical symptoms of the disease with a disease index of just 12 because of the inoculation of strain HAAG3-15, while 27 (90%) infected plantlets had a disease index of 45 in the F treatment. In addition, the other two treatments N (inoculated no microorganism) and S (inoculated only HAAG3-15) exhibited no symptoms of disease and still stayed healthy (Figure not shown). Moreover, the application of strain HAAG3-15 could also significantly increase the shoot fresh weight and height of cucumber (*p* < 0.05); the average shoot fresh weight and height of cucumber (4.62 g, 12.55 cm) in treatment S (inoculated only HAAG3-15) were greater than those in treatment N (no microorganism; 4.06 g, 11.76 cm), which clearly indicated that strain HAAG3-15 could promote the growth of cucumber seedlings. The employment of strain HAAG3-15 could also markedly reduce the impact of *F. oxysporum* f. sp*. cucumerinum* on cucumber shoot fresh weight and height (*p* < 0.05). For treatment F (inoculated only *F. oxysporum* f. sp*. cucumerinum*), the average shoot fresh weight and height were 3.16 g and 10.32 cm, respectively, which were lower than those for treatment F + S (inoculated both HAAG3-15 and *F. oxysporum* f. sp*. cucumerinum*; 3.95 g, 11.58 cm).

### 3.6. Structure Elucidation of the Antifungal Compound

The antifungal compound was separated based on the antifungal activity-guided method from the extraction of fermentation broth of strain HAAG3-15, and compound 1 was obtained as its active constituent. Then, compound 1 was identified as Azalomycin B (Figure 9) based on the spectral data (Appendix A) and literature values [72].

The antifungal activities of compound 1 against F. oxysporum f. sp. Cucumerinum, Corynespora cassiicola, Setosphaeriaturcica turcicaf, Colletotrichum orbiculare, Alternaria solani, Helminthosporium maydis, Sphacelotheca reiliana, Sclerotinia sclerotiorum, Phytophthora sojae, and Rhizoctonia solani were determined in vitro. The compound showed significant antifungal activity against F. oxysporum f. sp. cucumerinum and also exhibited certain antifungal activities against the nine other fungi. Therefore, Azalomycin B was identified as the main antifungal component produced by strain HAAG3-15.

## 4. Discussion

Microorganisms related to plant roots are vital for plant growth and health, and they are considered to be the second genome of the plant. When plants are attacked by pathogens, the diversity and community structure of plant-associated microbes may be changed [48,49,73]. Here we studied the differences in root-associated endophytic actinobacterial community composition and antifungal activity between *Fusarium* wilt diseased and healthy cucumber under natural greenhouse field conditions in China using culture-independent and culture-dependent analysis, and we screened actinomycetes for potential biological control of *Fusarium* wilt of cucumber. Microbiome biodiversity is known as a driver of plant health. Abundant bacterial flora predetermines the future plant health [49,74]. In this study, culture-independent analysis suggested that the healthy cucumber roots had higher actinobacteria richness and abundance than the diseased plants. This result was similar to the previous study showing that tomato plant resistance to infection with *Ralstonia solanacearum* depended on more abundance and diversity of rhizospheric bacteria than diseased plants [49]. Moreover, the relative abundance of phylum Actinobacteria in the healthy samples was more significant than in diseased samples. It is known that Actinobacteria can produce various metabolites with important potential application in the agriculture, food, and pharmaceutical industries [75,76], such as antibiotics, enzymes, enzyme inhibitors, vitamins, and so on. The reduction of the relative abundance of *Actinobacteria* may have had a positive effect on the *Fusarium oxysporum* f. sp. *cucumerinum* growth as a result of weakened pathogen suppression via antibiosis.

In addition, culture-dependent analysis was also performed using five isolation media to isolate strains from the cucumber roots. In total, 173 endophytic actinomycetes colonies were isolated from the healthy cucumber roots, and 90 endophytic actinomycetes colonies originated from the diseased cucumber roots (nearby the healthy cucumber collected from the cucumber continuous cropping greenhouse), which indicated that the culturable endophytic actinomycetes in the healthy cucumber roots were more abundant than in the diseased cucumber roots (Figure 4). The result was in agreement with the culture-independent analysis. To obtain actinobacteria with potential biological control of *Fusarium* wilt of cucumber, all strains were selected for testing its antifungal activity against *F. oxysporum* f. sp. *cucumerinum*. Results showed that there were seven strains with antifungal activity against *F. oxysporum* f. sp. *cucumerinum* in healthy cucumber roots, but only one strain in diseased cucumber roots with weak antifungal activity. In previous studies, several biocontrol and plant growth-promoting endophytes were isolated from infected plants [57,77,78] or healthy plants [79,80,81]. Our results seem to be different from previous observations, and the healthy cucumber root nearby the diseased plant contained more abundant microbes, as well as more actinomycetes with antifungal activities. Among the eight antagonistic strains, isolate HAAG3-15 in healthy cucumber root was found to be best as it had the strongest antifungal activity against *F. oxysporum* f. sp. *cucumerinum* and also exhibited broad-spectrum antifungal activity. Moreover, biocontrol of cucumber *Fusarium* wilt showed that strain HAAG3-15 had an obvious effect in terms of disease prevention and growth promotion on cucumber seedlings in greenhouse assay.

It was reported that biocontrol of the *Fusarium* wilt pathogens illustrated the use of suppressive soils and antagonistic bacteria to inhibit the propagation of germination and penetration growth by the pathogen [82]. However, a rapid decline in the size of populations of active cells to ineffective levels was achieved following introduction into soil, due to the hostility of the soil environment to incoming microbes [83]. However, endophytes are not subject to competition from soil microbes, and they colonize in the plant tissue. They have the ability to penetrate plant cells, stimulating the plant defense response and producing antifungal metabolites in situ. Strain HAAG3-15 was screened from the cucumber root, belonging to the group of endophytic actinomycetes, which would not affect the structure of the actinomycetes in the root of cucumber. If the endophytic actinomycetes were introduced into the cucumber seedlings at the breeding stage, they would become the principal parts of the microbial flora in the cucumber plant at the time of transplanting and could protect their host plant from *F*. *oxysporum* f. sp. *cucumerinum*. This is a promising prospect for biological control of *Fusarium* wilt of cucumber. It provides a new method for the prevention and cure of *F*. *oxysporum* f. sp. *cucumerinum* in agricultural production. Therefore, the endophytic strain HAAG3-15 as a biological control agent against *Fusarium* wilt of cucumber has great potential application in organic agriculture.

In addition, we also screened and identified antifungal components from strain HAAG3-15 and a bioactive compound, Azalomycin B, was obtained. Azalomycin, as a broad-spectrum antibiotic, is widely applicable [84,85,86,87,88]. Previous studies showed that Azalomycin has a good inhibitory effect on Gram-positive bacteria, parasites, and fungi, and the application prospect of Azalomycin in agricultural production was also presented [35,89,90,91]. Azalomycin B isolated in the current study also showed significant antifungal activity against *F. oxysporum* f. sp. *cucumerinum*, as also previously reported. Further research is needed to confirm the efficacy of the active compound against cucumber *Fusarium* wilt under greenhouse conditions.

Above all, the result of the present study indicated that a pathogen-prevalent environment, such as for healthy roots nearby infected plants, is also a good source for isolating biocontrol and plant growth-promoting endophytes, and the endophytic actinomycete strain HAAG3-15 has potential as a biocontrol agent against *F. oxysporum* f. sp. *cucumerinum*.

## 5. Conclusions

In conclusion, healthy cucumber roots had higher actinobacteria richness and abundance than diseased cucumber roots based on culture-independent and culture-dependent analysis. This suggested that the healthy root nearby the infected plant is a good source for isolating biocontrol and plant growth-promoting actinobacteria endophytes. In addition, strain HAAG3-15 showed stronger antifungal activity against *F. oxysporum* f. sp. *cucumerinum* than seven other strains, with an obvious effect in terms of disease prevention and growth promotion on cucumber seedlings in the greenhouse assay. Its excellent broad-spectrum antifungal activities suggest that it could be a potential candidate for the development of a potential biocontrol agent against *F. oxysporum* f. sp. *cucumerinum* used in organic agriculture.

## Figures and Tables

**Figure 1 microorganisms-08-00236-f001:**
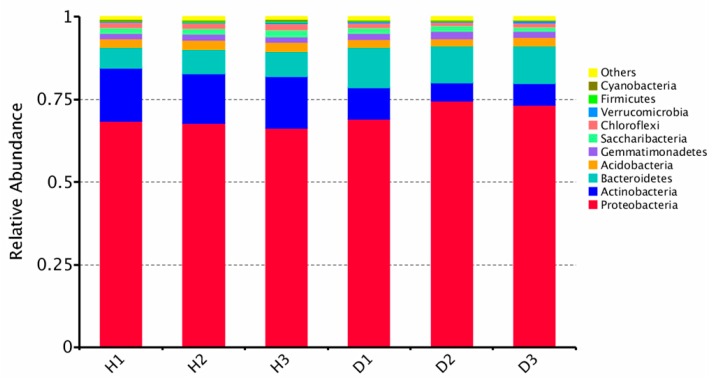
Analysis of culture-independent endophytic communities at phylum level in the cucumber roots. H1–3, healthy samples; D1–3, diseased samples.

**Figure 2 microorganisms-08-00236-f002:**
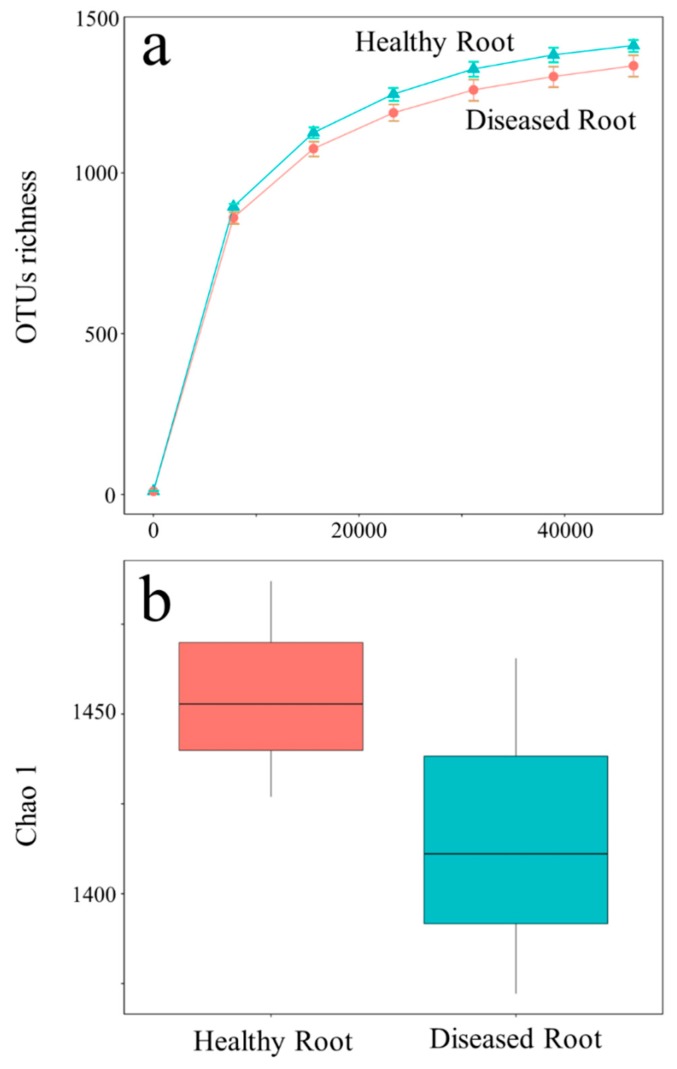
The rarefaction curve (**a**) and Chao 1 (**b**) α-diversity of the healthy and diseased cucumber roots.

**Figure 3 microorganisms-08-00236-f003:**
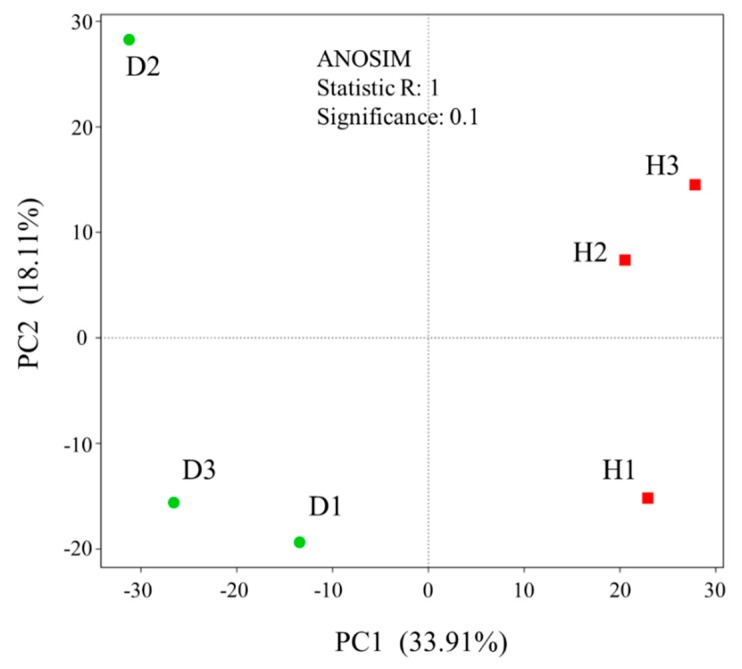
Principal component analysis (PCA) of bacterial community beta-diversity based on Bray–Curtis dissimilarity among all samples of the healthy and diseased roots.

**Figure 4 microorganisms-08-00236-f004:**
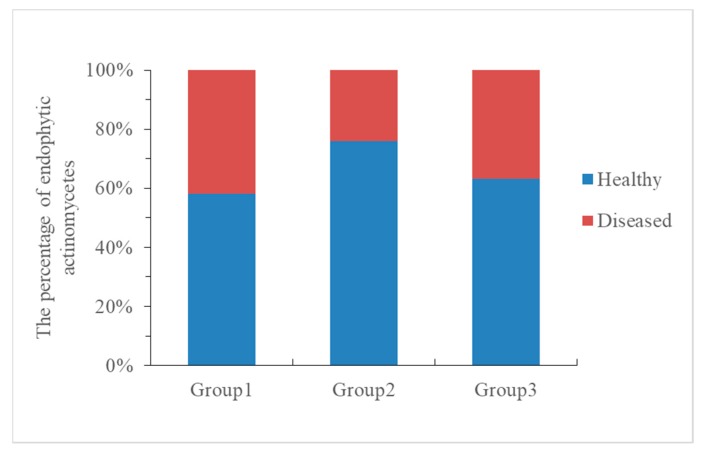
The percentage of endophytic actinomycetes in Group1 (H1-D1), Group2 (H2-D2), and Group3 (H3-D3).

**Figure 5 microorganisms-08-00236-f005:**
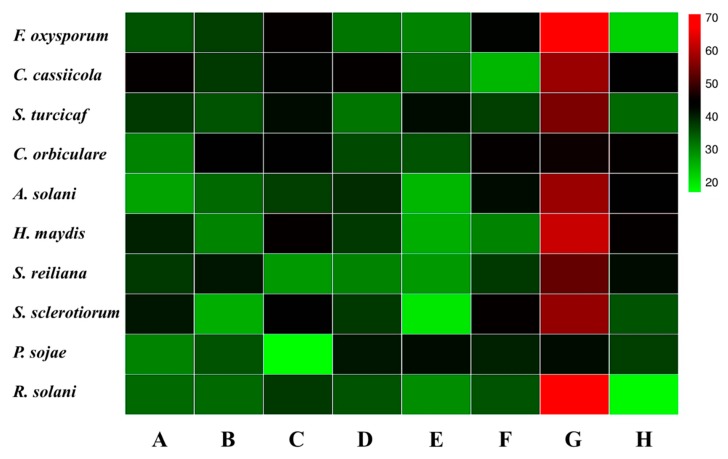
Eight isolates showed antagonism to *Fusarium oxysporum* f. sp*. cucumerinum* and nine other pathogenic fungi. **A**, HGS1-1; **B**, HGS2-18; **C**, HGS3-17; **D**, HAAG3-4; **E**, HAAG3-8; **F**, HCPA2-26; **G**, HAAG3-15; **H**, DCPA1-15. The color of the checkerboard represents the inhibition rate of fungal mycelial growth (%) of A–H. The legend on the right side represents the color corresponding to the different inhibition rates.

**Figure 6 microorganisms-08-00236-f006:**
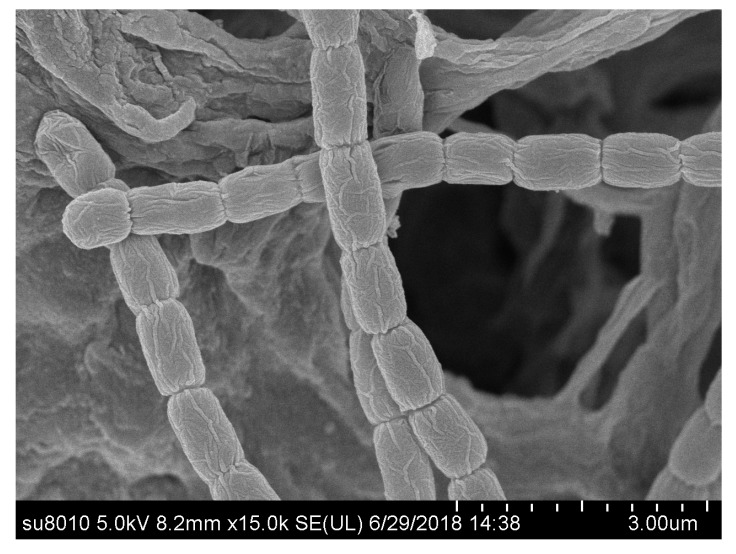
Scanning electron micrograph of strain HAAG3-15 grown on International *Streptomyces* Project (ISP) 3 medium for three weeks at 28 °C.

**Figure 7 microorganisms-08-00236-f007:**
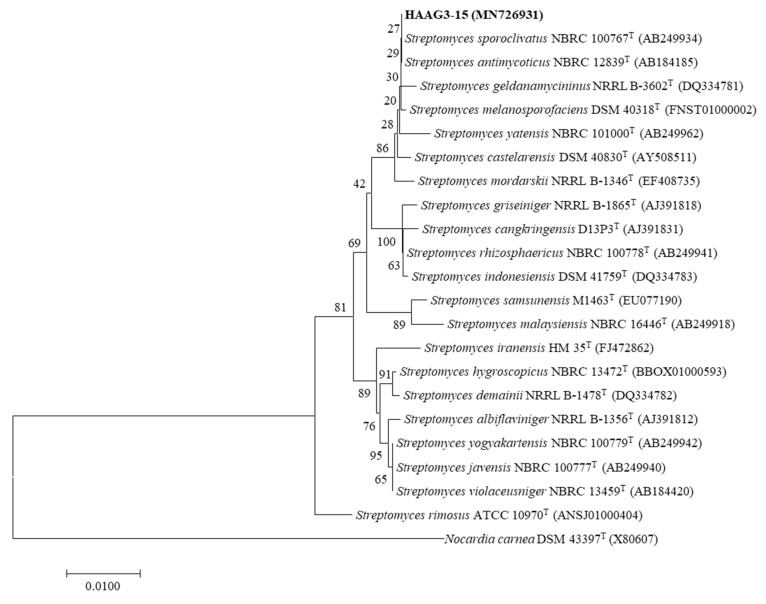
Neighbor-joining phylogenetic tree, based on almost-complete 16S ribosomal RNA (rRNA) gene sequences, showing the phylogenetic relationships of strain HAAG3-15 and the closest strains within the genus *Streptomyces*. *Nocardia carnea* DSM43397^T^ was used as the outgroup. Bar, 0.0100 substitutions per nucleotide position.

**Figure 8 microorganisms-08-00236-f008:**
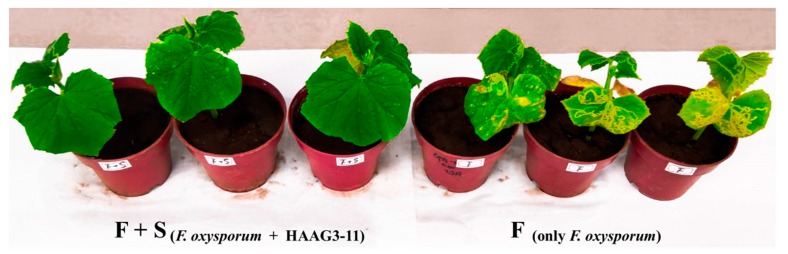
Biocontrol assay of inoculation with *F. oxysporum* f. sp*. cucumerinum* and strain HAAG3-15 on cucumber seedlings in greenhouse. Inoculation with *F. oxysporum* f. sp*. cucumerinum* and HAAG3-15 (left, F + S), and only inoculated *F. oxysporum* f. sp*. cucumerinum* (right, F).

**Figure 9 microorganisms-08-00236-f009:**
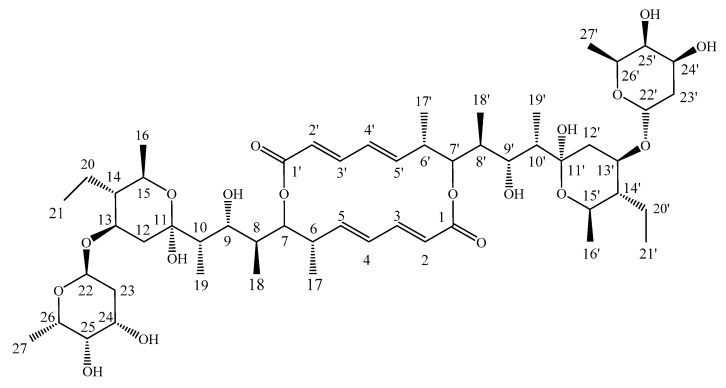
Structure of Azalomycin B. Structural elucidation of compound 1: Azalomycin B, C_54_H_88_O_18_, white microcrystals. UV (MeOH) λ_max_ 252nm; electrospray ionization (ESI) MS *m*/*z*: 1047 [M + Na]^+^; ^1^H NMR (600 MHz, MeOD) *δ*_H_ 6.93 (1H, dd, *J* = 15.2, 11.2 Hz, H-3), 6.17 (1H, dd, *J* = 15.0, 11.2 Hz, H-4), 5.74 (1H, d, *J* = 15.4 Hz, H-2), 5.67 (1H, dd, *J* = 15.1, 9.9 Hz, H-5), 5.04 (2H, m, H-7, H-22), 4.02 (1H, d, *J* = 9.7 Hz, H-13), 3.94 (2H, m, H-24, H-26), 3.90 (2H, m, H-9, H-15), 3.53 (1H, m, H-25), 2.58 (1H, m, H-6), 2.34 (1H, dd, *J* = 12.1, 4.5 Hz, H-12), 1.95 (1H, m, H-8), 1.94 (1H, dd, *J* = 12.3, 3.9 Hz, H-23), 1.72 (1H, m, H-10), 1.66 (1H, m, H-20), 1.62 (1H, dd, *J* = 12.6, 4.7 Hz, H-23), 1.47 (1H, m, H-20), 1.20 (3H, d, *J* = 6.6 Hz, H-27), 1.16 (1H, m, H-16), 1.13 (3H, d, *J* = 6.0 Hz, H-16), 1.12 (1H, m, H-12), 1.05 (3H, d, *J* = 6.6 Hz, H-17), 0.97 (3H, d, *J* = 7.1 Hz, H-19), 0.87 (6H, t, *J* = 6.2 Hz, H-21, H-18). ^13^C NMR (150 MHz, MeOD) *δ*_C_ 170.4 (C-1), 146.9 (C-3), 146.1 (C-5), 132.7 (C-4), 122.6 (C-2), 100.9 (C-11), 94.8 (C-22), 78.2 (C-7), 72.4 (C-25), 71.8 (C-9), 70.9 (C-13), 68.2 (C-15), 68.1 (C-26), 67.0 (C-24), 49.8 (C-14), 44.0 (C-10), 42.7 (C-6), 38.9 (C-12), 37.8 (C-8), 33.7 (C-23), 20.3 (C-20), 19.5 (C-16), 17.3 (C-27), 15.8 (C-17), 9.6 (C-18), 9.5 (C-21), 7.1 (C-19).

**Table 1 microorganisms-08-00236-t001:** Height, shoot fresh weight, disease index (DI), and number of infected plantlets with four treatments in greenhouse biocontrol assay.

Treatments	Height (cm)	Shoot Fresh Weight (g)	Disease Index	Infected Plantlets
F	10.32 ± 0.52 ^c^	3.16 ± 0.36 ^c^	45 ± 3.8 ^a^	27(90%)
F + S	11.58 ± 0.63 ^b^	3.95 ± 0.18 ^b^	12 ± 2.2 ^b^	10(30%)
N	11.76 ± 0.46 ^b^	4.06 ± 0.27 ^b^	0	0
S	12.55 ± 0.32 ^a^	4.62 ± 0.15 ^a^	0	0

Average shoot fresh weight and height of 15 plantlets for each treatment (mean ± SD). Different letters in the same column indicate significant differences (*p* < 0.05). Cucumber plants grown in soil containing F, the spore suspension of *F. oxysporum* f. sp. *cucumerinum* (2 mL of 4–5 × 10^4^ CFU/mL); F + S, the spore suspension of *F. oxysporum* f. sp. *cucumerinum* (2 mL of 4–5 × 10^4^ CFU/mL) and the spore suspension of strain HAAG3-15 (2 mL of 4–5 × 10^6^ CFU/mL); N, no microorganism (2 mL of sterile tap water); and **S**, the spore suspension of HAAG3-15 (2 mL of 4–5 × 10^6^ CFU/mL).

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
