# Peer review of "Community Structures and Antifungal Activity of Root-Associated Endophytic Actinobacteria in Healthy and Diseased Cucumber Plants and Streptomyces sp. HAAG3-15 as a Promising Biocontrol Agent"

_microorganisms, 2020, doi:10.3390/microorganisms8020236_

Round 1

Reviewer 1 Report

Cao et al investigated and compared microbial diversity of healthy and unhealthy cucumber plants.  The authors found higher number of actinobacteria OTUs in healthy plant sample relative to the unhealthy pairs as well as isolated more actinomycetes from the healthy roots.  Isolates from cucumber roots (mostly from healthy roots) are able to inhibit the growth of pathogenic fungi, including F. oxysporum.  Application of a selected isolate to cucumber seedlings in the greenhouse is also performed in the study.  Antifungal activity-guided natural product isolation leads to the identification of a known antifungal compound, azalomycin B or elaiophylin.

The manuscript presents a study to discover biological agents for plant pathogen control.  However, the study presented here apparently does not reflect completely to the goal described.  Perhaps, the authors can justify and describe the unhealthy plants selected in the study, e.g. were the diseased plants Fusarium-infected cucumber plants?.  The title of manuscript also gives an impression (at least to me) that the study investigated the response of actinobacterial communities in the root to Fusarium infection, i.e. before vs. after infection, but the manuscript contents tell differently.  Rephrasing topical sentences throughout the manuscript is strongly suggested. 

While the intended application of this study is clearly described in the manuscript, current version requires major revisions to improve the quality and overall benefit to publishing this work.  Please see details below.  

Broad comments/questions:
1. The introduction section has covered almost all necessary aspects to the study.

Adding examples of microbial biocontrol agents (including Streptomyces) would be desirable.

2. It would be useful to include additional details in method section and figure legends.

What filter values were used in the quality control (Cutadapt line 120)? What do dots and oval lines represent in Figure 3? Are there any outgroups and bootstrap values for Figure 7?

3.Clarification would be required for the methodology

Line 151: How to do point-inoculation? Using spores harvested from the isolates cultivated for 7 days (line 147-148)? Antagonistic effects seem to be a result of antifungal compound diffusion on agar plates, suggesting the compounds being secreted. However, compound 1 was isolated from HAAG3-15 mycelial cake (obtained from liquid/submerged cultures, line 213).  How do the authors determine the antagonistic effects resulting from the same compound? Does azalomycin B antagonise the other 9 pathogenic fungi? Line 214: How to separate supernatant and wash water using HP20 resin? Figure 5 contains undescribed figure legends (e.g. values from 20 to 70). If the authors include a figure representing growth inhibition of fungal pathogen, it would be greatly useful to illustrate what colored tiles (checker board) represent. Were the 7 antagonists (line 269) from the three groups (H1-H3)? It is confusing as to why average of shoot height and weight is derived from 15 plantlets while infected plantlets seemed from the analysis of 30 plantlets The number of inoculants was fixed in the greenhouse experiments, does alteration of the amount/concentration affect the plant growth

4. While LC-MS and NMR data is presented in the main text, please include the chromatogram and NMR spectra in the manuscript. If it is in the supplemental information, please refer to the figure.

5. The authors showed the relative abundance of endophytic actinomycetes in Figure 4. Are the values significantly different? Please include the statistics as being done elsewhere in the manuscript. Similar issue with line 241: what is the significant level?

6. Some figures in the pdf form are blurry, e.g. error bars in Figure 2A are not clear

Specific/additional comments:

Common terminologies should be used in the manuscript, e.g Erlenmeyer flasks instead of triangular flasks (line 207), menstruum (line 215) is not typically used in the field Significant designation (a, b, c) in Table 1 should be in superscript form Line 437: stronger than what? What is the obvious effect written in line 438? The conclusion is strongly suggested for further editing. Abstract is also a part for language polishing, e.g. the first sentence refers microbes as the second genome – this seems to be unparalleled comparison Typos can be found in the manuscript, e.g. micorbes (line 81), punctuation (line 218)

Author Response

Dear reviewer,

Thank you for the valuable suggestions. In the following, we provide our itemized list of changes according to your suggestions and highlighted the changes in our manuscript.

Thank you very much for your kindness and help.

Sincerely yours,

Dr. Peng Cao and Prof. Junwei Zhao

The manuscript presents a study to discover biological agents for plant pathogen control. However, the study presented here apparently does not reflect completely to the goal described. Perhaps, the authors can justify and describe the unhealthy plants selected in the study, e.g. were the diseased plants Fusarium-infected cucumber plants? The title of manuscript also gives an impression (at least to me) that the study investigated the response of actinobacterial communities in the root to Fusarium infection, i.e. before vs. after infection, but the manuscript contents tell differently. Rephrasing topical sentences throughout the manuscript is strongly suggested.

Reply: Thank you very much for your valuable suggestions. We have revised and changed the title to ‘Community structures and antifungal activity of root-associated endophytic actinobacteria in healthy and diseased cucumber and Streptomyces sp. HAAG3-15 as a promising biocontrol agent’ and topical sentences to ‘The goal of this study is to characterize differences in root-associated endophytic actinobacterial community composition and antifungal activity between Fusarium wilt diseased and healthy cucumber and screen actinobacteria for potential biological control of Fusarium wilt of cucumber.’ Please see lines 2-5, 17-20, and 95-98.

Broad comments/questions:

The introduction section has covered almost all necessary aspects to the study. Adding examples of microbial biocontrol agents (including Streptomyces) would be desirable.

Reply: Thank you very much and we have added examples of microbial biocontrol agents (including Streptomyces). Please see lines 69-72 and 74-78.

It would be useful to include additional details in method section and figure legends. What filter values were used in the quality control (Cutadapt line 120)?

Reply: The raw data was filtered according to the software Cutadapt (Version 1.9.1) and removed low quality reads less than 17, and we have revised in the manuscript. Please see Lines 126-127.

What do dots and oval lines represent in Figure 3?

Reply: We have modified the Figure 3, and marked the points in the figure. Please see Figure 3.

Are there any outgroups and bootstrap values for Figure 7?

Reply: We have modified the Figure 7, and added outgroup and bootstrap in the figure. Please see Figure 7 and line 315.

Clarification would be required for the methodology.

Line 151: How to do point-inoculation? Using spores harvested from the isolates cultivated for 7 days (line 147-148)?

Reply: The strains were point-inoculated using inoculating needle and we have revised. Please see line 159. These isolates were streaked on ISP 3 medium and cultivated for 7 days and then we harvested spores of them. Please see line 154-155.

Antagonistic effects seem to be a result of antifungal compound diffusion on agar plates, suggesting the compounds being secreted. However, compound 1 was isolated from HAAG3-15 mycelial cake (obtained from liquid/submerged cultures, line 213).

Reply: The supernatant and mycelial cake wash water as well as MeOH extraction of mycelial cake were all harvested for separating the antifungal compound based on the antifungal activity-guided method. Please see lines 221-223.

How do the authors determine the antagonistic effects resulting from the same compound? Does azalomycin B antagonise the other 9 pathogenic fungi?

Reply: The antifungal compound was separated based on the antifungal activity-guided method from the extraction of fermentation broth of strain HAAG3-15 and compound 1 was obtained as its active constituent. The compound 1 showed significantly antifungal activity against F. oxysporum f. sp. cucumerinum and also exhibited certain antifungal activities against other 9 fungi. Therefore, the compound was identified as the main antifungal principle which produced by strain HAAG3-15. We have revised and please see lines 372-376.

Line 214: How to separate supernatant and wash water using HP20 resin?

Reply: The supernatant and wash water were subjected to a Diaion HP-20 resin column eluting with 95 % EtOH (5 L). We have revised and please see lines 221-222.

Figure 5 contains undescribed figure legends (e.g. values from 20 to 70). If the authors include a figure representing growth inhibition of fungal pathogen, it would be greatly useful to illustrate what colored tiles (checker board) represent.

Reply: Thank you very much and we have revised, please see lines 290-292.

Were the 7 antagonists (line 269) from the three groups (H1-H3)?

Reply: 7 antagonists were isolated from healthy cucumber roots (H1-H3). Please see lines 279-280.

It is confusing as to why average of shoot height and weight is derived from 15 plantlets while infected plantlets seemed from the analysis of 30 plantlets.

Reply: The disease symptoms of all cucumber seedlings per treatment were investigated in this study. Fifteen cucumber seedlings randomly harvested from the pots of each treatment were used to measure their shoot fresh weights and heights. Please see lines 205-208.

The number of inoculants was fixed in the greenhouse experiments, does alteration of the amount/concentration affect the plant growth.

Reply: In the preliminary experiments, we found that different concentration can affect the plant growth and disease incidence. And HAAG3-15 spore suspension with concentration 106 CFU ml-1 showed the best activity in vivo. Therefore, we choose this concentration in the greenhouse assay.

While LC-MS and NMR data is presented in the main text, please include the chromatogram and NMR spectra in the manuscript. If it is in the supplemental information, please refer to the figure.

Reply: We have referred to the figure S2 and figure S3 in the line 357. Please see line 357.

The authors showed the relative abundance of endophytic actinomycetes in Figure 4. Are the values significantly different? Please include the statistics as being done elsewhere in the manuscript. Similar issue with line 241: what is the significant level?

Reply: Yes the values significantly different. We have revised and added the significant level (P < 0.05). Please see lines 248 and 272.

Some figures in the pdf form are blurry, e.g. error bars in Figure 2A are not clear.

Reply: We have overstriking error bars and please see Figure 2A.

Specific/additional comments:

Common terminologies should be used in the manuscript, e.g Erlenmeyer flasks instead of triangular flasks (line 207).

Reply: We have revised and please see lines 215-216.

menstruum (line 215) is not typically used in the field.

Reply: We have revised and please see lines 222-223.

Significant designation (a, b, c) in Table 1 should be in superscript form.

Reply: We have revised in Table 1.

Line 437: stronger than what?

Reply: Strain HAAG3-15 showed stronger antifungal activity against F. oxysporum f. sp. Cucumerinum than other 7 strains. We have revised and please see lines 450-451.

What is the obvious effect written in line 438?

Reply: The obvious effect on disease prevention and growth promotion on cucumber seedlings in greenhouse assay. We have revised and please see lines 451-452.

The conclusion is strongly suggested for further editing.

Reply: Thank you very much and we have re-edited the conclusion. Please see lines 447-455.

Abstract is also a part for language polishing, e.g. the first sentence refers microbes as the second genome – this seems to be unparalleled comparison.

Reply: Thank you very much. It's a metaphor that “Microorganisms related to plant roots are vital for plant growth and health and considered to be the second genome of the plant. When the plant is attacked by plant-pathogen, the diversity and community structure of plant-associated microbes might be changed.” And this viewpoint has been put forward by others like Berendsen et al. and Wei et al. in their previous reports as follows:

Berendsen, R.L.; Pieterse, C.M.; Bakker P.A. The rhizosphere microbiome and plant health. Trends Plant Sci. 2012, 17(8), 478-486.

Wei, Z.; Hu, J.; Gu, Y.A.; Yin, S.X.; Xu, Y.C.; Jousset, A.; Shen, Q.R.; Friman, V.P. Ralstonia solanacearum pathogen disrupts bacterial rhizosphere microbiome during an invasion. Soil Biol. Biochem. 2018,118, 8-17.

Typos can be found in the manuscript, e.g. micorbes (line 81), punctuation (line 218).

Reply: We have revised and please see lines 88, 224-225, and 232.

Reviewer 2 Report

This is an excellent paper, covering an important subject. There are no amendments required.

Author Response

Thank you very much!

Reviewer 3 Report

The authors described response of endophytic actinobacterial communities in cucumber root to infection with Fusarium oxysporum f. sp. cucumerinum and Streptomyces sp. HAAG3-15 as a promising biocontrol agent. In these days when the need for alternative strategies is increasing such study is more than welcome. I've been stated some comments within the manuscript. However, I have several main worries. First of all, the authors highlight the diversity of communities but without going in details related to the differences on genera level between healthy and diseased plants. Also, it’s not perfectly clear how the authors made an isolation of active compounds, how they performed TLC test (within the results this part is missing) and obtained just only one compound. More clarifications need here. In addition, the first part of discussion has a lot of repetition of the mentioned results, and very poorly comparison with other study particularly related to culture-independent analysis. Better discussion parts are missing.

Author Response

Dear reviewer,

Thank you for the valuable suggestions. In the following, we provide our itemized list of changes according to your suggestions and highlighted the changes in our manuscript.

Thank you very much for your kindness and help.

Sincerely yours,

Dr. Peng Cao and Prof. Junwei Zhao

The authors described response of endophytic actinobacterial communities in cucumber root to infection with Fusarium oxysporum sp. cucumerinum and Streptomyces sp. HAAG3-15 as a promising biocontrol agent. In these days when the need for alternative strategies is increasing such study is more than welcome. I've been stated some comments within the manuscript. However, I have several main worries. First of all, the authors highlight the diversity of communities but without going in details related to the differences on genera level between healthy and diseased plants.

Reply: Thank you very much and we have added the differences of communities on genera level (Figure S1) between healthy and diseased plants. Please see Figure S1 and lines 249-251.

Also, it’s not perfectly clear how the authors made an isolation of active compounds, how they performed TLC test (within the results this part is missing) and obtained just only one compound. More clarifications need here.

Reply: The crude extract was resolved by a silica gel (100-200 mesh) column eluted with a stepwise gradient of CHCl3/MeOH mixtures with a growing polarity (100:0-50:50, v/v) to obtain three fractions (Fr.1-3) based on the TLC profiles, which was performed with solvent system of CHCl3/MeOH (9:1). F2 showed antifungal activity and was further purified by a Sephadex LH-20 gel column eluted with CHCl3/MeOH (1:1, v/v) and gave two fractions Fr.2-1 and Fr.2-2 refer to the TLC profiles. The Fr.2-1 showed antifungal activity and was further separated by semi-preparative HPLC. We have revised and please see lines 228-230.

The antifungal compound was separated based on the antifungal activity-guided method from the extraction of fermentation broth of strain HAAG3-15 and compound 1 was obtained as its active constituent. The compound 1 showed significantly antifungal activity against F. oxysporum f. sp. cucumerinum and also exhibited certain antifungal activities against other 9 fungi. Therefore, the compound was identified as the main antifungal principle which produced by strain HAAG3-15. We have revised and please see lines 372-376.

In addition, the first part of discussion has a lot of repetition of the mentioned results, and very poorly comparison with other study particularly related to culture-independent analysis. Better discussion parts are missing.

Reply: Thank you very much and we have revised the first part of discussion. Please see lines 385-391.

Round 2

Reviewer 1 Report

The authors have revised the manuscript substantially – Thank you very much for the effort!  This includes the clarity of research design and method as well as the suitability of the overall conclusion with the hope being to increase the accessibility of the study. Now the manuscript is in a better shape; however, there are still some missing information and minor editing are expected.  Please see below (reviewed per point):

The manuscript presents a study to discover biological agents for plant pathogen control. However, the study presented here apparently does not reflect completely to the goal described. Perhaps, the authors can justify and describe the unhealthy plants selected in the study, e.g. were the diseased plants Fusarium-infected cucumber plants? The title of manuscript also gives an impression (at least to me) that the study investigated the response of actinobacterial communities in the root to Fusarium infection, i.e. before vs. after infection, but the manuscript contents tell differently. Rephrasing topical sentences throughout the manuscript is strongly suggested.

Reply: Thank you very much for your valuable suggestions. We have revised and changed the title to ‘Community structures and antifungal activity of root-associated endophytic actinobacteria in healthy and diseased cucumber and Streptomyces sp. HAAG3-15 as a promising biocontrol agent’ and topical sentences to ‘The goal of this study is to characterize differences in root-associated endophytic actinobacterial community composition and antifungal activity between Fusarium wilt diseased and healthy cucumber and screen actinobacteria for potential biological control of Fusarium wilt of cucumber.’ Please see lines 2-5, 17-20, and 95-98.

Additional review: Suggestion for the title: add “plants” after “…healthy and diseased cucumber…”

Broad comments/questions:

Are there any outgroups and bootstrap values for Figure 7?

Reply: We have modified the Figure 7, and added outgroup and bootstrap in the figure. Please see Figure 7 and line 315.

Additional edit: Please match information provided in Figure 7 and its figure legend (e.g. bar for nucleotide substitution).

Clarification would be required for the methodology.

Antagonistic effects seem to be a result of antifungal compound diffusion on agar plates, suggesting the compounds being secreted. However, compound 1 was isolated from HAAG3-15 mycelial cake (obtained from liquid/submerged cultures, line 213).

Reply: The supernatant and mycelial cake wash water as well as MeOH extraction of mycelial cake were all harvested for separating the antifungal compound based on the antifungal activity-guided method. Please see lines 221-223.

Additional edit: Line 222 “…resin column and the column was washed with…”.  Please add the size of the column used.

Figure 5 contains undescribed figure legends (e.g. values from 20 to 70). If the authors include a figure representing growth inhibition of fungal pathogen, it would be greatly useful to illustrate what colored tiles (checker board) represent.

Reply: Thank you very much and we have revised, please see lines 290-292.

Additional review: please add a figure that shows growth inhibition of fungal pathogen, e.g. a plate cultivated with strain HAAG3-15 and its inhibitory effect.

While LC-MS and NMR data is presented in the main text, please include the chromatogram and NMR spectra in the manuscript. If it is in the supplemental information, please refer to the figure.

Reply: We have referred to the figure S2 and figure S3 in the line 357. Please see line 357.

Additional review: please add the LC-MS figure.

Specific/additional comments:

Line 437: stronger than what?

Reply: Strain HAAG3-15 showed stronger antifungal activity against F. oxysporum f. sp. Cucumerinum than other 7 strains. We have revised and please see lines 450-451.

Additional edits: Cucumerinum should not be capitalized.  Please check the entire manuscript.

Author Response

Dear reviewer,

Thank you for the valuable suggestions. In the following, we provide our itemized list of changes according to your suggestions and highlighted the changes in our manuscript.

Thank you very much for your kindness and help.

Sincerely yours,

Dr. Peng Cao and Prof. Junwei Zhao

Additional review: Suggestion for the title: add “plants” after “…healthy and diseased cucumber…”

Reply: Thank you very much for your valuable suggestions. We have revised. Please see line 4.

Broad comments/questions:

Additional edit: Please match information provided in Figure 7 and its figure legend (e.g. bar for nucleotide substitution).

Reply: Thank you very much and we have revised. Please see line 317.

Clarification would be required for the methodology.

Additional edit: Line 222 “…resin column and the column was washed with…”. Please add the size of the column used.

Reply: We have added. Please see line 222.

Additional review: please add a figure that shows growth inhibition of fungal pathogen, e.g. a plate cultivated with strain HAAG3-15 and its inhibitory effect.

Reply: Thank you very much and we have revised, please see Figure S2.

Additional review: please add the LC-MS figure..

Reply: We have added ESI-MS figure in the supplementary file as Figure S5. Please see lines 237-238 and Figure S5.

Specific/additional comments:

Additional edits: Cucumerinum should not be capitalized. Please check the entire manuscript.

Reply: We have revised and please see line 453.